# *Prunus spinosa* Extract Loaded in Biomimetic Nanoparticles Evokes In Vitro Anti-Inflammatory and Wound Healing Activities

**DOI:** 10.3390/nano11010036

**Published:** 2020-12-25

**Authors:** Mattia Tiboni, Sofia Coppari, Luca Casettari, Michele Guescini, Mariastella Colomba, Daniele Fraternale, Andrea Gorassini, Giancarlo Verardo, Seeram Ramakrishna, Loretta Guidi, Barbara Di Giacomo, Michele Mari, Roberto Molinaro, Maria Cristina Albertini

**Affiliations:** 1Department of Biomolecular Sciences, University of Urbino Carlo Bo, 61029 Urbino (PU), Italy; mattia.tiboni@uniurb.it (M.T.); s.coppari3@campus.uniurb.it (S.C.); luca.casettari@uniurb.it (L.C.); michele.guescini@uniurb.it (M.G.); mariastella.colomba@uniurb.it (M.C.); daniele.fraternale@uniurb.it (D.F.); loretta.guidi@uniurb.it (L.G.); barbara.digiacomo@uniurb.it (B.D.G.); michele.mari@uniurb.it (M.M.); 2Department of Humanities and Cultural Heritage, University of Udine, 33100 Udine, Italy; andrea.gorassini@uniud.it; 3Department of Agricultural, Food, Environmental and Animal Sciences, University of Udine, 33100 Udine, Italy; giancarlo.verardo@uniud.it; 4Center for Nanofibers and Nanotechnology, National University of Singapore, Singapore 119077, Singapore; mpesr@nus.edu.sg; 5IRCCS Ospedale San Raffaele srl, 20132 Milan, Italy

**Keywords:** biomimicry, lipid nanoparticles, drug delivery systems, phenolic compounds, scratch assay, micro-RNA, leukosome

## Abstract

*Prunus spinosa* fruits (PSF) contain different phenolic compounds showing antioxidant and anti-inflammatory activities. Innovative drug delivery systems such as biomimetic nanoparticles could improve the activity of PSF extract by promoting (i) the protection of payload into the lipidic bilayer, (ii) increased accumulation to the diseased tissue due to specific targeting properties, (iii) improved biocompatibility, (iv) low toxicity and increased bioavailability. Using membrane proteins extracted from human monocyte cell line THP-1 cells and a mixture of phospholipids, we formulated two types of PSF-extract-loaded biomimetic vesicles differing from each other for the presence of either 1,2-dioleoyl-sn-glycero-3-phosphocholine (DOPC) or 1,2-dioleoyl-sn-glycero-3-phospho-(1′-rac-glycerol) (DOPG). The biological activity of free extract (PSF), compared to both types of extract-loaded vesicles (PSF-DOPCs and PSF-DOPGs) and empty vesicles (DOPCs and DOPGs), was evaluated in vitro on HUVEC cells. PSF-DOPCs showed preferential incorporation of the extract. When enriched into the nanovesicles, the extract showed a significantly increased anti-inflammatory activity, and a pronounced wound-healing effect (with PSF-DOPCs more efficient than PSF-DOPGs) compared to free PSF. This innovative drug delivery system, combining nutraceutical active ingredients into a biomimetic formulation, represents a possible adjuvant therapy for the treatment of wound healing. This nanoplatform could be useful for the encapsulation/enrichment of other nutraceutical products with short stability and low bioavailability.

## 1. Introduction

The wound-healing process involves different phases: hemostasis, inflammation, tissue proliferation, and tissue remodeling [1]. Tissue injury causes the immediate onset of the acute inflammatory response, which is essential to provide growth factor and chemokines signals that orchestrate cell movements and tissue remodeling necessary for repair. Successful repair after tissue injury requires inflammatory resolution that starts with the production of anti-inflammatory cytokines and downregulation of proinflammatory mediators (Figure 1). An excessive or prolonged inflammatory phase results in increased tissue injury and poor healing. The process depends on the persistence of inflammatory cells, the consequent generation of proinflammatory cytokines (Interleukin 1 and 6, tumor necrosis factor α), and on the increase of reactive oxygen species (ROS), which can lead to direct damage of cells or extracellular matrix molecules [2].

Bioactive phytochemical constituents such as phenolic compounds can promote beneficial effects on wound healing through anti-inflammatory and antioxidant activities. Moreover, they can modulate one or more phases of the wound-healing process [3]. They assist in wound healing, mainly with inhibitory effects on the production and activity of inflammatory mediators.

Among plant products, *Prunus spinosa* Linnaeus (*P. spinosa*) fruits have been extensively described for their biological properties and beneficial effects [4,5]. It is a perennial deciduous plant growing as a shrub in wild, uncultivated areas of Europe, temperate regions of Asia, and Mediterranean countries. In particular, chemical composition, antioxidant and anti-inflammatory properties of *P. spinosa* fruit ethanol extract have been recently reported [6]. The most representative molecules in the extract are phenolic compounds such as anthocyanins, which represent the most abundant antioxidants in the diet. This high content of active molecules and the resulting nutraceutical potential could prevent chronic diseases, including chronic complications in wound closure [7,8].

Polyphenol-rich plant extracts can interfere with the production of cytokines and, therefore, can offer an important alternative for the treatment of inflammatory diseases [9]. The main disadvantage of using natural extracts is their low bioavailability. Effective concentrations of these substances are unlikely to be found in the bloodstream and therefore hardly exert anti-inflammatory activity in their site of action.

Nanomedicine arose as an innovative tool to overcome these drawbacks [10]. Indeed, drug delivery systems (DDS), such as lipid vesicles [11,12] and polymeric nanoparticles [13], showed several advantages in enhancing the pharmacokinetic properties of their payload by promoting, for example, its protection from enzymatic or chemical degradation [14], preferential targeting towards specific cells or tissues [15], improved biocompatibility, low toxicity and increased bioavailability [16].

Among the different nanomedicine approaches, recently, biomimetic nanoparticles were developed as the last generation DDS endowed with unexpected biological properties [17,18,19,20]. Biomimetic approaches include mimicking leukocytes [21], red blood cells [22], platelets [23], and cancer cells [24]. These approaches provided unique functionalities by incorporating cellular coating, synthesizing cell-derived nanovesicles, or mimicking the physical properties of cells to create novel nanoparticles. Biomimetic carriers showed innate biological features and intrinsic functionalities typical of the donor cell source [18]. Among them, leukosomes, i.e., leukocyte-like nanovesicles, showed prolonged circulation and preferential targeting of inflamed vasculature [20,25,26,27].

Here, we enriched biomimetic vesicles with *P. spinosa* fruit ethanolic extract (hereafter-termed PSF) and evaluated whether its “encapsulation” provided improved biological properties in an in vitro setting, recapitulating the wound-healing process. From a pharmaceutical standpoint, PSF has been incorporated in two different biomimetic nanovesicles formulations, termed leukosomes, differing from each other for the presence of either 1,2-dioleoyl-*sn*-glycero-3-phosphocholine (DOPC), a neutral phospholipid, or 1,2-dioleoyl-*sn*-glycero-3-phospho-(1′-*rac*-glycerol) (DOPG), an anionic phospholipid, in the lipid mixture (hereafter reported as DOPCs and DOPGs, respectively).

The object of the present work is to evaluate and characterize the effective incorporation of the active molecules present in PSF from both a qualitative and a quantitative standpoint and to evaluate the ability of the formulations to reduce the inflammation process and improve the wound-healing repair.

## 2. Materials and Methods

### 2.1. Chemicals and Materials

1,2-dioleoyl-*sn*-glycero-3-phosphocholine (DOPC), 1,2-dioleoyl-*sn*-glycero-3-phospho-(1′-*rac*-glycerol) (DOPG), 1,2-distearoyl-*sn*-glycero-3-phosphocholine (DSPC), 1,2-Dipalmitoyl-*sn*-glycero-3-phosphocholine (DPPC), and cholesterol (CHOL) were purchased from Avanti Polar Lipids (Alabaster, AL, USA). Chloroform (CHCl_3_), methanol (MeOH), pure ethanol (EtOH), and formic acid (HCOOH) for High Performance Liquid Cromatography coupled with mass spectrometry (HPLC-MS), chlorogenic acid, phloridzin, gallic acid, cyanidin chloride, uranyl acetate were purchased from Sigma-Aldrich (Milan, Italy). Quercetin-3-O-galactoside was obtained from ExtraSynthese (Lyon, France). Quercetin-3-O-arabinoside and quercetin-3-O-rhamnoside were purchased from Carbosynth (Berkshire, UK). Milli-Q^®^ grade water was produced by the Elgastat UHQ-PS system (ELGA, High Wycombe Bucks, UK). Solid phase extraction (SPE) columns ISOLUTE C18, 1 g, 6 mL were from Biotage (Milan, Italy).

### 2.2. Fruits Collection and P. spinosa Extract Preparation

*P. spinosa* L. (Also known as blackthorn, Rosaceae family) was identified by Prof. D. Fraternale (University of Urbino). Voucher samples were deposited in the herbarium of Urbino University Botanical garden: code “Pp125”. Ripe fruits were collected in November 2019 from plants around Urbino (PU, Marche, Italy), GPS coordinates N43°40′46.512″ E12°31′42.291″ (350 m above sea level) and, immediately after harvesting, frozen and stored at −20 °C until use. For the extraction, 50 g of fruit pulp were homogenized in an Osterizer for 3 min in 50 mL of 70% acidified aqueous ethanol solution (70 mL ethanol/29.9 mL distilled H_2_O/0.1 mL HCl 36%). The homogenized solution was filtered with a Buchner filter, and the filtrate was centrifuged at 10,000 g for 15 min. The supernatant obtained was collected, while the pellet and the residues of the first extraction were subjected again to the initial extraction conditions. Both supernatants were combined and concentrated under vacuum at 37 °C using a rotary evaporator. In the end, the final product was aliquoted in centrifuge tubes and dried using a Savant concentrator (Thermo Scientific, San Jose, CA, USA). The dried samples were stored at −20 °C until use.

### 2.3. Formulation of Biomimetic Nanovesicles

Biomimetic vesicles were manufactured as previously reported [21,28]. Briefly, two different mixtures of phosphocholine-based phospholipids and cholesterol: DPPC:DOPC:DSPC:CHOL (named DOPCs) or DPPC:DOPG:DSPC:CHOL (named DOPGs) at a molar ratio of 5:3:1:1, were dissolved in chloroform into a round-bottomed flask and then, the solvent was evaporated using a rotary evaporator (Laborota 4000, Heidolph, Germany) to form a lipidic film according to the well-established thin layer evaporation method. A precise amount of freeze-dried PSF ethanolic extract was added to the lipid phase (1:6.67 weight ratio) to prepare PSF extract-loaded nanoparticles (named as PSF-DOPCs and PSF-DOPGs). Films were then hydrated with a water dispersion of human monocyte (THP-1, ATCC) membrane proteins (1:300 protein-to-lipid ratio) to assemble the biomimetic vesicles. The lipidic suspensions obtained were extruded ten times through 200 nm pore-size cellulose acetate membranes (Avanti Polar Lipid, Alabama, AL, USA) at 45 °C to obtain smaller and more homogeneous vesicles.

### 2.4. Physicochemical Characterization of Biomimetic Nanovesicles

The prepared vesicles were characterized with a Nanosizer ZS (Malvern Instrument, Worcestershire, UK) using the dynamic light scattering (DLS) technique to measure the average particle size (Z-average) and their dimensional homogeneity, i.e., polydispersity index (PDI). Moreover, the Z-potential was measured to determine the surface charge of the plain (empty) and loaded biomimetic vesicles.

Both the formulated biomimetic suspensions were then ultracentrifuged at 100,000 g for 1 h and freeze-dried (Lio5P, 5Pascal, Milano, Italy) to characterize the incorporated PSF active molecules by HPLC-DAD-ESI-MS^n^ (see Appendix A).

### 2.5. Cell Culture

Human umbilical vein endothelial cells (HUVECs) used in this study are derived from multiple donors, purchased from Life Technologies Corporation (Gibco™ C01510C, 1 × 10^6^ cells). The HUVECs were cultured in the growth medium of Gibco™ Medium 200 (Life Technologies Corporation, Grand Island, NY, USA). This basal medium requires the addition of Gibco™ low serum growth supplement kit (LSGS Kit) (Life Technologies Corporation, Grand Island, NY, USA), and the final concentrations of the components in the integrated medium are: fetal bovine serum (2% *v/v*), hydrocortisone (1 μg/mL), growth factor human epidermal (10 ng/mL), basic fibroblast growth factor (3 ng/mL) and heparin (10 μg/mL). Cultures were passaged on, reaching 80% confluence, using trypsin–EDTA solution (Sigma-Aldrich, St. Louis, MO, USA). The experiments were performed in the third culture passage (P3).

### 2.6. In Vitro Wound Healing Assay

HUVECs were seeded in T25 tissue culture flasks, incubated at 37 °C and 5% CO_2_ and allowed to grow to confluence as a monolayer. The cell monolayer was subjected to a mechanical scratch wound, horizontal along the flask, using the tip of a sterile pipette. Subsequently, the medium of each flask was removed, and the cells were washed with PBS solution. Diluting with fresh medium, the cells (200,000/mL) were treated with 80 µg/mL of free PSF, empty vesicles (DOPCs and DOPGs), and 80 µg/mL PSF-loaded vesicles (PSF-DOPCs and PSF-DOPGs). The untreated cells were used as a control and the incubation time was 48 h for all conditions. After the treatment, to evaluate cell viability, the cells were detached from the flask, and the cell suspension was mixed with trypan blue. Then, the cells that took up (dead cells) or excluded (vital cells) the dye were evaluated [29]. The images were obtained from the same lesion area immediately after scratching (t_0_) and then after 24 and 48 h using a phase-contrast microscope (Olympus Ix51 10X objective). To evaluate wound closure, images were analyzed through the ImageJ software to select the total area of the wound region. The percentage of wound closure was calculated using Equation (1).
[(wound area t_0_ − wound area t)/wound area t_0_] × 100(1)

The experiments were conducted in triplicate.

### 2.7. RT–qPCR Analysis

The analysis was performed on the HUVECs used for the wound-healing assay. The RT–qPCR protocol includes total RNA isolation, which was performed through the Total RNA isolation kit (Norgen Biotek, Thorold, ON, USA) following the manufacturer’s instructions. The concentration and purity of the RNA were determined using a NanoDrop ND 1000 spectrophotometer (Thermo Scientific, San Jose, CA, USA). RNA was stored at -80 °C until use.

The cDNA was synthesized from the isolated RNA using the High-Capacity cDNA reverse transcription kit (Applied Biosystems, Foster City, CA, USA). RT–qPCR was performed with the SYBR Green PCR master mix (Applied Biosystems, Foster City, CA, USA) on ABI Prism 7500 real-time PCR system (Applied Biosystems, Foster City, CA, USA). TATA-binding protein (TBP) has been used as an endogenous control to determine relative mRNA expression. Primer sequences used were as follow:

VCAM-1 (F: ACAGAAGAAGTGGCCCTCCAT-R: TGGCATCCGTCAGGAAGTG); IL-6 (F: AGGGCTCTTCGGCAAATGTA-R: GAAGGAATGCCCATTAACAACAA); IRAK-1 (F: CAGACAGGGAAGGGAAACATTTT-R: CATGAAACCTGACTTGCTTCTGAA); TBP (F: TGCACAGGAGCCAAGAGTGA-R: CACATCACAGCTCCCCACCA).

For miRNA analysis, human miR-126 and human miR-146a were quantified by RT–qPCR using TaqMan MicroRNA assay (Applied Biosystems, Foster City, CA, USA) according to the manufacturer’s guidelines. The RT–qPCR data were standardized to RNU44 (reference miRNA).

Product specificity was examined by dissociation curve analysis. Results were calculated using the delta-delta Ct method (2^−ΔΔCt^) and were expressed as fold change related to untreated control (CTRL).

### 2.8. Statistical Analysis

The two-tailed paired Student’s *t*-test was used for the miR-126, VCAM-1, miR-146a, IRAK-1, IL-6 analyses. The results were considered significant at the level of *p* < 0.05. All the experiments were conducted in triplicate. Linear regression analysis was used to evaluate the wound-healing dose-dependent capacity, where R-squared (R^2^) measures how close the data are to the fitted regression line (the higher the R^2^ to 100%, the better the model fits the data).

## 3. Results

### 3.1. Physicochemical Characterization of Empty and Extract-Loaded Vesicles

We undertook to characterize the contribution of two different lipid mixtures having the DPPC:DSPC:CHOL as lipid backbone by adding either DOPC or DOPG to obtain an almost neutral (DOPCs) or negatively charged formulation (DOPGs).

Empty and loaded DOPCs and DOPGs, assembled with the technique developed by Molinaro et al. [21], were characterized from a physicochemical standpoint by measuring their average size (PDI, and surface charge (Z-potential) (Figure 2).

DOPCs and DOPGs revealed an average size of 171 nm and 135 nm, respectively, a narrow size distribution (PDI below 0.13) and a surface charge of −13 mV and −35 mV, respectively. PSF incorporation did not significantly affect size distribution, while we could observe some differences regarding the average diameter and surface charge (Figure 1). PSF incorporation induced a significant reduction of vesicle size for the DOPCs (171 vs. 140 nm before and after PSF-loading, respectively), while the effect was only minimal for the DOPGs (135 vs. 125 nm before and after PSF-loading, respectively). On the contrary, DOPGs showed a significant reduction of the surface charge after PSF incorporation (−35 vs. −28 mV before and after PSF-loading, respectively), while a very slight difference could be observed for the DOPCs (−13 vs. −11 mV before and after PSF-loading, respectively).

Taken together, these findings reveal a different effect of PSF incorporation in the two formulations, which was predictable considering both the heterogeneity of PSF composition and the net difference between DOPC and DOPG, which confer distinct physicochemical properties to the vesicles.

HPLC chromatograms acquired at 280 nm of purified samples of free PSF, PSF-DOPCs and PSF-DOPGs are reported in Appendix A, and the characterization of the main phenolic compounds is reported in Appendix A.

As reported in Table 1, the dominant classes of phenolic compounds present in PSF are hydroxycinnamic acid derivatives (44.4%), anthocyanins (32.7%), and flavonoid derivatives (21.1%), which is in accordance with literature data [30,31,32].

We observed preferential incorporation within PSF-DOPCs than PSF-DOPGs for hydroxycinnamic acid derivatives (542 vs. 125 µg/g, respectively), hydroxybenzoic acid derivatives (60 vs. 51 µg/g, respectively), and flavonoid derivatives (687 vs. 235 µg/g, respectively) (Table 1). Overall, the amount of total phenolic compounds detected in PSF-DOPCs was about three times higher than the amount found in PSF-DOPGs (1339 µg/g vs. 464 µg/g, respectively).

As mentioned above, phenolic compounds are biologically relevant molecules with an active role in the control of oxidative stress and consequently of the inflammatory response [33,34]. Among the most abundant polyphenols present in *P. spinosa* extract, HPLC–DAD/ESI–MS^n^ detected 3-O-caffeoylquinic acid (2112 µg/g), peonidin 3-O-rutinoside (650 µg/g), cyanidin 3-O-rutinoside (639 µg/g), cyanidin 3-O-glucoside (490 µg/g), quercetin galactoside (367 µg/g), peonidin 3-O-glucoside (315 µg/g), quercetin hexoside (240 µg/g), and 3-O-feruloylquinic acid (218 µg/g).

Furthermore, quercetin xyloside was mostly loaded by PSF-DOPCs than by PSF-DOPGs (29 vs. 4 µg/g, respectively), as well as quercetin 3-O-rhamnoside, which was undetected in PSF-DOPGs.

### 3.2. Wound Healing Repair Activity of PSF-Loaded Vesicles

We investigated the wound-healing properties of *P. spinosa* fruit extract at different concentrations (20 to 80 μg/mL) on HUVECs up to 48 h of incubation (Figure 3).

We found that the percentage of wound closure after free PSF treatment showed a concentration-dependent effect. Linear regression analysis showed R^2^ = 84% considering 20–80 µg/mL of free PSF, and R^2^ = 98% considering 20–40 µg/mL of free PSF. On this basis, we deduced that the wound-healing capacity of the extract is saturated at 80 µg/mL.

We further evaluated whether incorporated PSF had similar activity. To this end, wound-healing assay was performed to the monolayer of HUVECs in different experimental conditions using: the free extract at the most effective concentration (80 μg/mL); the extract-loaded vesicles (PSF-DOPCs and PSF-DOPGs) and the plain vesicles (DOPCs and DOPGs). 80 μg/mL PSF-loaded nanovesicles were employed. After the treatment, no cytotoxicity was noticed for any formulation with the trypan blue exclusion assay. The free PSF and PSF-DOPCs groups showed a significant wound-healing repair activity compared to the control (Figure 4). More specifically, PSF-DOPCs showed increased wound-healing repair even related to free PSF alone.

### 3.3. Anti-Inflammatory Activity of PSF-Loaded Vesicles

To dissect the biological mechanisms at the basis of the wound-healing activity above reported, we evaluated the anti-inflammatory ability of the free PSF, either free or incorporated. To this end, we measured the expression of the inflammation markers miR-146a, IRAK-1, and IL-6 after 48 h of treatment.

As indicated in Figure 5, the relative expression of the proinflammatory cytokine IL-6 is significantly lower in the groups treated with free PSF and PSF-DOPCs than the control. Interestingly, both empty DOPCs and DOPGs revealed an intrinsic anti-inflammatory activity, thus confirming previous reports [28], while we could not observe any significant effect for loaded PSF-DOPGs. Furthermore, we observed a decreased IRAK-1 and increased miR-146a expressions for free PSF and PSF-DOPCs groups, with the latter having a more pronounced effect.

To support previous results, the expression of miR-126 and its target VCAM-1 were analyzed for all the experimental conditions (Figure 6). Furthermore, in this case, in free PSF and PSF-DOPCs groups, decreased VCAM-1 and increased miR-126 expression levels were observed. Treatment with empty DOPCs lowered the expression of VCAM-1, thus confirming the intrinsic anti-inflammatory properties observed for these biomimetic nanovesicles.

## 4. Discussion

In this work, we have demonstrated how biomimetic nanoparticles can improve the nutraceutical anti-inflammatory properties of *P. spinosa* extract, derived from a wild fruit widely present in the Mediterranean area. Phenolic compounds contained in the PSF extract demonstrated different nutraceutical effects such as antioxidant and anti-inflammatory activities [35] that can be enhanced when paired with the active targeting to inflamed endothelium, as carefully characterized in previous works [21,36].

Taking advantage of the well-established thin layer evaporation method, purified membrane proteins extracted from human monocyte cell line THP-1 were assembled with synthetic choline-based phospholipids to mimic the physiologic composition of the plasmalemma [21,37]. This manufacturing process allowed for the production of biomimetic vesicles that combine the high surface complexity typical of cell-derived nanoparticles with the pharmaceutical versatility typical of liposomes, such as the ability to carry, transport, and release different kinds of payloads, an elevated and standardized yield of the manufacturing process and a stable, biocompatible and non-immunogenic final product. Moreover, this manufacturing protocol does not require chemical synthesis or complex purification steps to reach the final biomimetic product.

The prepared vesicles presented an optimal dimension and narrow size distribution with effective incorporation of the *P. spinosa* extract. In particular, PSF-DOPCs presented a number of phenolic compounds three times higher than PSF-DOPGs. We observed the same trend for hydroxycinnamic acid and flavonoid derivatives (four and three times higher for PSF-DOPCs compared to PSF-DOPGs). More specifically, PSF-DOPCs preferentially incorporated quercetin xyloside (present in free extract), unlike PSF-DOPGs. These differences could be attributed to electronic interactions between the active compounds present in the extract and the utilized phospholipids. Our findings reveal that the neutral DOPC allowed for higher incorporation of those compounds compared to the DOPG, which has a net negative charge that most likely hindered their loading.

As this kind of biomimetic vesicles presented an effective active targeting to inflamed endothelium as reported already in literature [28], the prepared nanovesicle formulations were studied on in vitro inflammatory conditions evaluating the wound-healing properties on endothelial HUVEC cells. Analyzing the percentage of wound closure, treatment with PSF-DOPCs showed a significantly higher wound-healing activity compared to all other experimental conditions (Figure 7).

In the wound-healing process, immediately after the wound, an acute inflammation mediated by cytokines and growth factors (e.g., IL-1, IL-6, TNF-α) occurs [2]. Among these, interleukin 6 (IL-6) plays an important role in wound healing. Indeed, the progression towards complete wound healing involves the downregulation of proinflammatory cytokines, essential for correct repair. TLR4 is an important regulator of inflammation [38]. The TLR4 signaling pathway, through the IRAK-1 mediator, promotes the activation of the NF-κB transcriptional complex, which is responsible for the production of proinflammatory cytokines, including IL-6. The miR-146a also regulates TLR4 pathway activation through a negative feedback loop. Hence, during a proinflammatory condition, miR-146a is downregulated and allows the expression of IRAK-1 and, consequently, IL-6.

We evaluated the expression of markers directly related to the inflammation and wound-healing processes such as IL-6, miR-146a, IRAK-1, miR-126, and VCAM-1. MicroRNA-126 is also responsible for regulating cell adhesion molecule VCAM-1 during inflammation [6].

The expression level of IL-6 was significantly decreased during free *P. spinosa* extract and PSF-DOPCs treatment. Moreover, in the same experimental groups, the limited expression of IRAK-1 and the high expression of miR-146a demonstrate the activation of the negative miR-146a-mediated feedback loop mechanism that modulates the TLR4 signaling pathway and the consequent inflammatory profile, thus confirming a significant anti-inflammatory activity of the active compounds of the extract. In line with recent experimental evidence showing that miR-126 is upregulated during wound healing, thus promoting cell migration and proliferation [39], while the expression of VCAM-1 is decreased [40], our findings showed that PSF-DOPCs significantly increased the expression level of miR-126 and decreased expression of its target VCAM-1. Authors speculate that PSF-DOPCs increased the intracellular bioavailability of some components of PSF extract, thus modulating the expression of proinflammatory markers, also through the antioxidant activity.

On the other hand, PSF-DOPGs did not significantly modulate the molecules associated with the TLR4 pathway, as for the empty DOPCs and DOPGs. However, the latter lowered the level of IL-6 associated with inflammation, probably due to the intrinsic anti-inflammatory capacity of the biomimetic nanovesicles [19]. The wound-healing effect of PSF-DOPGs was considerably lower compared to the free PSF extract and PSF-DOPCs probably due, as we report below, to (i) less efficient incorporation; and (ii) differences in active molecules incorporated (as shown by HPLC). In particular, PSF-DOPCs contained a higher amount of hydroxycinnamic acids (Table 1—Peaks no. 1–5, 7) and flavonoids including quercetin derivatives (Table 1—Peaks no. 6, 9, 17–24) that are known as wound-healing modulators [41]. Furthermore, molecules highly loaded into PSF-DOPCs such as caffeic acid and quercetin derivatives (quercetin xyloside) have already been explored in the literature showing an antioxidant [42] and anti-inflammatory effect through downregulation of the TLR-4 signaling pathway [43,44]. On this basis, our findings seem to confirm that these are the components of the PSF extract most directly involved in determining the observed experimental results.

As for the mechanism of action, the authors hypothesize that the improved anti-inflammatory activity of PSF extract results from their enhanced delivery by biomimetic nanovesicles. We have previously reported how leukosomes show a preferential targeting towards inflamed HUVEC [21], which results in an increased intracellular release of their payload, thus modulating the inflammatory cascade resulting in anti-inflammatory activity. Since there are no data on how long the vesicles remain loaded before emptying yet, the experiments have always been performed with the vesicles freshly loaded in order to reasonably exclude that part of the payload was released into the culture medium before vesicles engaged HUVECs. Obviously, further in-depth analyses are needed to better clarify this point. In addition, a possible synergistic effect between the anti-inflammatory action of some components of the extract and the intrinsic anti-inflammatory capacity of the nanovesicles should not be ruled out.

Authors speculate that the incorporation of PSF within DOPGs (negatively charged) induced a perturbation of the system, probably destabilizing the correct incorporation of membrane proteins inside the lipid bilayer. This effect is most likely a consequence of the electrostatic interactions occurring between the DOPG and the compounds present in the ethanolic extract, in particular the anthocyanins, which are positively charged and the acidic components, in their negatively ionized form, could be responsible for destabilizing the membrane of PSF-DOPG through electrostatic repulsive forces. This validates the incorporation efficiency findings reported in Table 1, where DOPCs showed increased loading properties compared to the DOPGs. Further physicochemical analyses will be performed to understand the dynamics of these interactions, as well as their effect on membrane proteins incorporation.

## 5. Conclusions

In this work, we demonstrated that PSF-DOPCs were more efficient than free *P. spinosa* fruit extract in promoting the wound-healing process through an anti-inflammatory activity, indicating an additive effect between the properties of the natural molecules and the drug delivery potential of the biomimetic nanovesicles (Figure 7). In addition, we highlighted how the chemical properties of the payload must be considered in view of a possible perturbation of the system with a consequent loss of activity of the complex biomimetic surface of leukosomes. This innovative biomimetic nanosystem, which combines nutraceutical active ingredients, purified membrane proteins, and fully biocompatible synthetic lipids, represents a possible effective adjuvant therapy for the treatment of wound healing. Biomimetic nanovesicles platform could be useful for the encapsulation/incorporation of other nutraceutical products, whose pharmacological properties in vivo are hindered by their poor chemical stability and low bioavailability.

## Figures and Tables

**Figure 1 nanomaterials-11-00036-f001:**
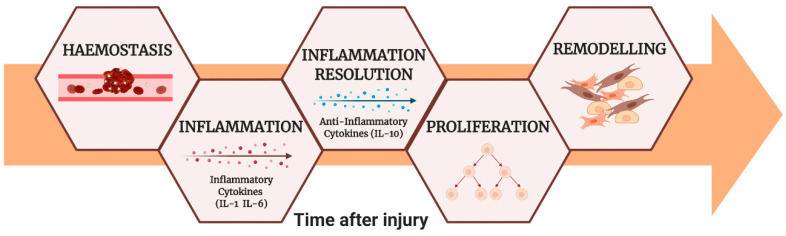
The wound-healing phases.

**Figure 2 nanomaterials-11-00036-f002:**
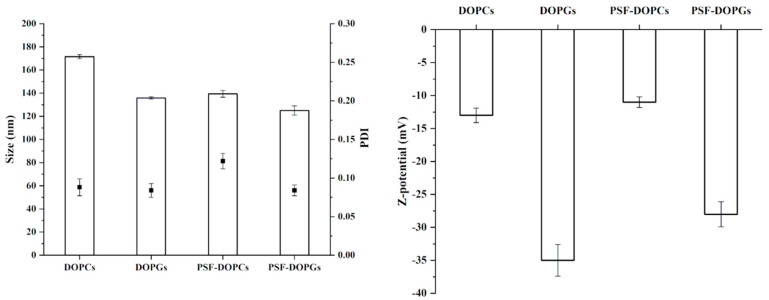
Characterization of formulated vesicles by Z-average size, polydispersity index (PDI), and Z-potential.

**Figure 3 nanomaterials-11-00036-f003:**
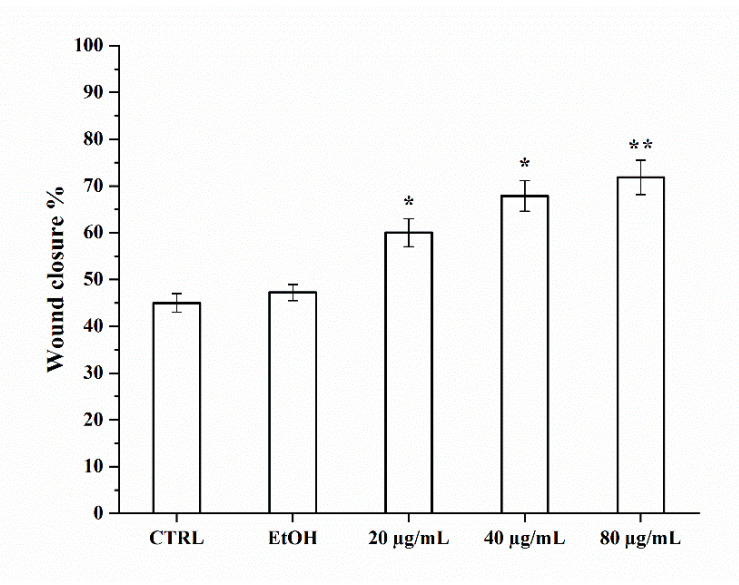
Wound-healing repair after *P. spinosa* extract (20–80 μg/mL) and its vehicle alone (EtOH) treatment. The graph shows the percentage of wound closure compared to the untreated control (CTRL) after 48 h. The values reported are the mean ± SD of three independent experiments. * *p* < 0.05, ** *p* < 0.001 vs. CTRL (Student’s *t*-test).

**Figure 4 nanomaterials-11-00036-f004:**
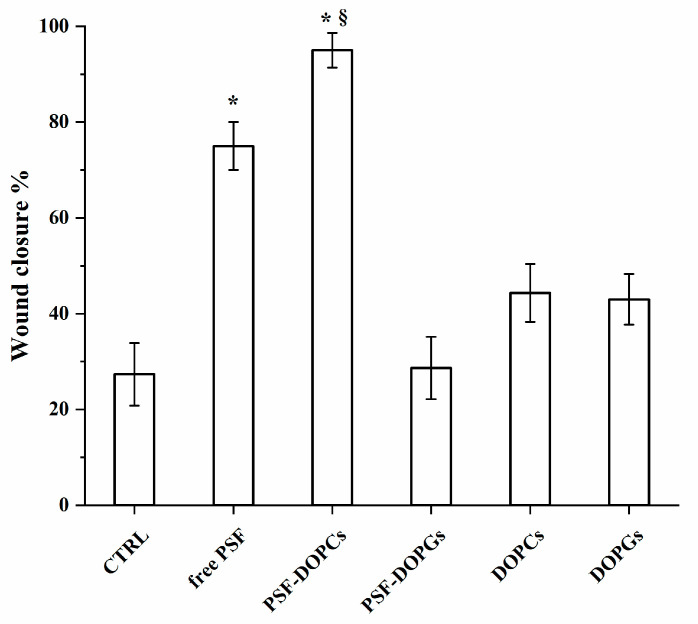
Wound-healing repair after free *P. spinosa* fruit extract (free PSF), extract-loaded vesicles (PSF-DOPCs and PSF-DOPGs) and empty vesicles (DOPCs and DOPGs) treatment. The graph shows the percentage of wound closure compared to untreated control (CTRL) after 48 h. The values reported are the mean ± SD of three independent experiments. * *p* < 0.05 vs. CTRL (Student’s *t*-test). ^§^
*p* < 0.05 vs. *P. spinosa* extract.

**Figure 5 nanomaterials-11-00036-f005:**
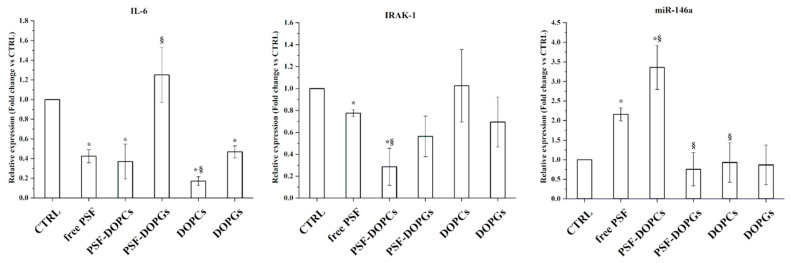
Effect of free *P. spinosa* fruit extract (free PSF), incorporated extract (PSF-DOPCs and PSF-DOPGs), and empty vesicles (DOPCs and DOPGs) on the IL-6, IRAK-1, and miR-146a expressions. RT–qPCR values are reported as fold induction related to CTRL. The values reported are the mean ± SD of three independent experiments. * *p* < 0.05 vs. CTRL (Student’s *t*-test). ^§^
*p* < 0.05 vs. *P. spinosa* extract.

**Figure 6 nanomaterials-11-00036-f006:**
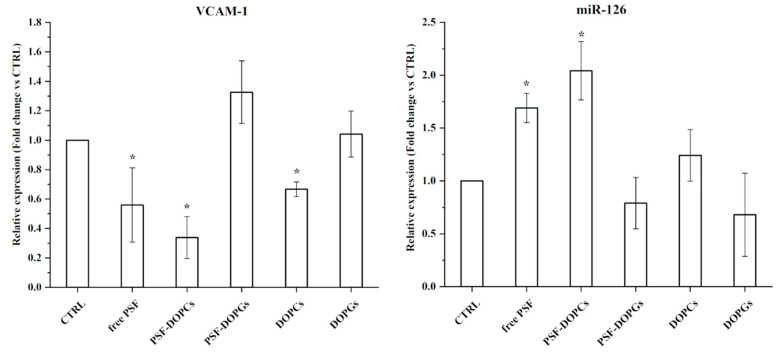
Effect of free *P. spinosa* fruit extract (free PSF), incorporated extract (PSF-DOPCs and PSF-DOPGs) and empty vesicles (DOPCs and DOPGs) on the VCAM-1 and miR-126 (cell adhesion molecules target). RT–qPCR values are reported as fold induction related to CTRL. Two-tailed paired Student’s *t*-test: * = *p* < 0.05.

**Figure 7 nanomaterials-11-00036-f007:**
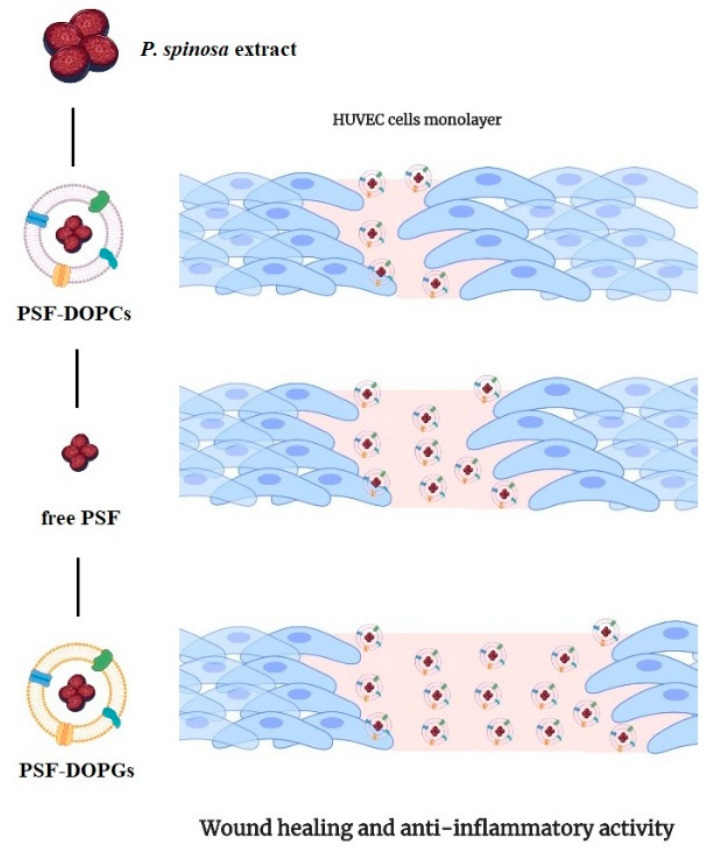
Free *P. spinosa* extract (free PSF) and loaded biomimetic formulations (PSF-DOPCs and PSF-DOPGs) acting on wound healing. As shown, PSF-DOPCs are the most efficient in wound closure, followed by free PSF and, finally, PSF-DOPGs, which are the least performing.

**Table 1 nanomaterials-11-00036-t001:** The total concentration of phenolic compounds in *P. spinosa* fruit ethanolic extract (free PSF) and in extract-loaded vesicles (PSF-DOPCs and PSF-DOPGs).

Peak No	Compound	Free PSF (µg/g)	PSF-DOPCs (µg/g)	PSF-DOPGs (µg/g)
1	3-O-caffeoylquinic acid	2112.0 ± 23.3	418.4 ± 13.0	94.0 ± 1.3
2	3-O-p-cumaroylquinic acid	70.0 ± 3.9	9.6 ± 0.1	3.5 ± 0.2
3	Chlorogenic acid dehydrodimer	75.8 ± 0.8	11.2 ± 0.1	2.7 ± 0.3
4	3-O-feruloylquinic acid	218.4 ± 1.4	37.2 ± 0.3	9.3 ± 0.3
5	4-O-caffeoylquinic acid	187.3 ± 3.9	41.2 ± 0.3	8.7 ± 0.1
7	Chlorogenic acid dehydrodimer	184.5 ± 2.3	24.8 ± 0.1	6.9 ± 0.2
	**Total hydroxycinnamic acid derivatives**	**2847.9 ± 23.5**	**542.3 ± 12.5**	**125.1 ± 1.5**
8	Cyanidin 3-O-glucoside	490.5 ± 6.3	26.4 ± 0.1	13.3 ± 0.1
10	Cyanidin 3-O-rutinoside	638.9 ± 1.9	11.2 ± 0.1	29.4 ± 0.2
11	Peonidin 3-O-glucoside	315.0 ± 4.4	4.6 ± 0.1	3.9 ± 0.2
12	Peonidin 3-O-rutinoside	650.3 ± 0.8	7.4 ± 0.1	7.2 ± 0.1
	**Total anthocyanins**	**2094.8 ± 11.8**	**49.7 ± 0.1**	**53.8 ± 0.1**
15	Ellagic acid derivative	40.6 ± 1.5	19.7 ± 0.2	10.1 ± 0.1
16	4-(vanilloyloxy)-2,6,6-trimethylcyclohexene-1-carboxylic acid	74.6 ± 2.2	40.6 ± 0.7	40.6 ± 0.2
	**Total hydroxybenzoic acid derivatives**	**115.1 ± 0.7**	**60.3 ± 2.3**	**50.7 ± 0.3**
6	Apigenin pentoside	27.4 ± 0.1	9.9 ± 0.1	3.9 ± 0.1
9	Apigenin pentoside isomer	28.1 ± 1.6	6.3 ± 0.1	4.3 ± 0.1
17	Quercetin hexoside	240.7 ± 6.4	64.6 ± 0.1	54.5 ± 0.1
18	Quercetin 3-O-hexoside-O-pentoside	145.3 ± 1.7	78.0 ± 0.1	24.6 ± 0.2
19	Rutin	168.9 ± 1.2	75.8 ± 0.2	23.9 ± 0.1
20	Quercetin galactoside	366.8 ± 4.8	215.7 ± 0.1	73.7 ± 0.2
21	Quercetin xyloside	29.5 ± 0.4	29.0 ± 0.1	4.1 ± 0.1
22	Quercetin arabinoside	107.7 ± 0.4	78.9 ± 0.2	22.5 ± 0.1
23	Quercetin pentoside	193.1 ± 1.1	108.0 ± 0.1	23.2 ± 0.1
24	Quercetin 3-O-rhamnoside	43.6 ± 0.3	20.8 ± 0.1	n.d.
	**Total flavonoid derivatives**	**1351.0 ± 8.4**	**687.1 ± 13.0**	**234.7 ± 0.1**
	**Total phenolic compounds**	**6408.9 ± 4.1**	**1339.4 ± 13.1**	**464.3 ± 1.1**

## Data Availability

The data presented in this study are available on request from the corresponding author.

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
