# Peer review of "Prunus spinosa Extract Loaded in Biomimetic Nanoparticles Evokes In Vitro Anti-Inflammatory and Wound Healing Activities"

_nanomaterials, 2020, doi:10.3390/nano11010036_

Round 1

Reviewer 1 Report

This is an interesting manuscript. Noteworthy, some controls are missed. To better discern the benefit of encapsulated extracts of Prunus spinosa on nanoparticles in respect to the extract more information is required.

Figure 3. It is not clear if this graph is showing the wound healing capacity dependence upon concentration or amount of extract. Please correct.

In addition and as shown in figure 3, the data show a practical linear correlation upon extract amount, but this might not be true based on the selected type of graph. Please plot this graph as points instead of bars to show if the wound healing capacity of the extracts is saturated at 80ug of extract. In case the graph does not show saturation, please increase the data points above 80ug/mL.

The same experiments should be performed with the loaded extracts in the nanoparticles. This experiment will allow proving if there is any improvement when cells are incubated with respect to cells incubated with the extracts alone.

The rest of the experiments where the gene expresión of inflammatory markers is analyzed show the same type of problem. It is not clear if the effect is due to cell incubation with the extract loaded on nanoparticles or due to their reléase from them. For Figures 5 and 6, please include the experiments showing the effect of the extracts without encapsulation. This will allow discerning if the effect is due to the complexes of the extracts and see any differences or improvement or Benefit of cell treatment with the extracts with respect to the encapsulated extracts.

A better integration discusión is required. What is the meaning of observed changes in the inflammatory biomarkers and how this data correlates with the different treatments. 

Minor points

Figure 2, 5, and 6. The legend of the X legend is unreadable. Please increase the letter size. Moreover, for the case of figure 5 and 6, please include the legend for the Y axes ad increase de size o the X

Author Response

The authors are very thankful to the reviewer for the quality of the suggestions that helped us to improve the overall quality of the manuscript. Please, find below the answers to reviewer’s comments.

This is an interesting manuscript. Noteworthy, some controls are missed. To better discern the benefit of encapsulated extracts of Prunus spinosa on nanoparticles in respect to the extract more information is required.

To better highlight the controls used, we changed the acronyms all over the text (including Figures and Tables) in order to better discern the effect of the encapsulated extract of Prunus spinosa in respect to the free extract.

Figure 3. It is not clear if this graph is showing the wound healing capacity dependence upon concentration or amount of extract. Please correct.

The wound healing capacity is concentration dependent. The X-axis legend of Figure 3 was corrected by adding µg/mL to be more clear.

In addition and as shown in figure 3, the data show a practical linear correlation upon extract amount, but this might not be true based on the selected type of graph. Please plot this graph as points instead of bars to show if the wound healing capacity of the extracts is saturated at 80ug of extract. In case the graph does not show saturation, please increase the data points above 80ug/mL.

We do agree with the reviewer on the usefulness of linear regression analysis to evaluate the saturation of wound healing capacity. Therefore, grateful for this suggestion, we added this analysis in M&M (see lines:199-200 ) and Results (see lines:256-259 ) sections. Nevertheless, in line with other previous studies, and aiming at a uniformity of representation, we keep the bars instead that the points graph.

The same experiments should be performed with the loaded extracts in the nanoparticles. This experiment will allow proving if there is any improvement when cells are incubated with respect to cells incubated with the extracts alone.

We thank the reviewer for the idea which is certainly interesting, but difficult to implement by in vitro cell experiments. In fact, due to technical issues related to the methods of extracting the active compounds of the extract from the vesicles, there is a risk of altering the secondary metabolites and maintaining compounds that could interfere with or even damage the cells.

The rest of the experiments where the gene expresión of inflammatory markers is analyzed show the same type of problem. It is not clear if the effect is due to cell incubation with the extract loaded on nanoparticles or due to their reléase from them. For Figures 5 and 6, please include the experiments showing the effect of the extracts without encapsulation. This will allow discerning if the effect is due to the complexes of the extracts and see any differences or improvement or Benefit of cell treatment with the extracts with respect to the encapsulated extracts.

This issue has been discussed in the revised version (see lines: 376-386)

Figure 5 and 6: we changed the acronyms in order to better discern the effect of the encapsulated extract of Prunus spinosa in respect to the free extract.

A better integration discusión is required. What is the meaning of observed changes in the inflammatory biomarkers and how this data correlates with the different treatments.

The meaning of observed changes in the inflammatory biomarkers has been better described in the new integrated discussion (see lines: 350-362).

Minor points

Figure 2, 5, and 6. The legend of the X legend is unreadable. Please increase the letter size. Moreover, for the case of figure 5 and 6, please include the legend for the Y axes ad increase de size o the X

All figures have been corrected as suggested.

Reviewer 2 Report

The manuscript “ Pronus Spinosa extract loaded in biomimetic nanoparticles evokes in vitro anti-inflammatory and wound healing activities” providing information on comparing two different lipid mixture systems (DOPC and DOPG)  for efficient encapsulation of Prunus Spinosa extract with anti-inflammatory activity and in vitro wound healing property.

The value of the phenolic compound is evaluated in two different lipid -protein biomimetic structures in comparison with free freeze-dried extract. Then, all samples are examined for in vitro wound healing properties and effect on the expression of some inflammation markers. Totally, the results have provided a good comparison  between two different  biomimetic structure  for encapsulation of phenolic compounds, however, the abstract and discussion  sections must be written with more precision, some corrections are needed  and some questions must be answered, including:

  • The abstract section must be re-written with more explanation on two different examined vesicles and also the obtained results.
  • Abstract (Line 28) : the “ good physicochemical properties” is an unclear concept. Please explain what do you mean with “good”.
  • Abstract(Line 29): the content of the phenolic compound in which formulation?
  • Line 62-63, Is not related to the subject of the article.
  • I suggest using better keywords instead of Micro-RNA or DDS, maybe anti-inflammatory and phenolic compounds.
  • Lines 69, 86 explain DDS, DOPG and DOPC when you use these for the first time. Also mention the abbreviation name of compounds in lines 94 and 95.
  • Why HUVECs were used for considering the wound healing potential of compounds?
  • For in vitro wound healing test please identify the number of cells/ml that were exposed to 80 µg of freeze-dried extract or extract loaded vesicles.
  • Line 340, indicate the reference.
  • Have you checked the toxicity of two different vesicles against the examined cells?

According to results, free extract and PS-DOPC leukosome exhibited higher wound healing activity and a significant effect on the expression of inflammatory markers. But these effects are not obvious for PS-DOPG leukosome. How the authors can explain this difference. can it be related to the content of phenolic compounds in the different systems as presented in table 1. If yes, how you can explain more efficiency of PS-DOPC in comparison with free extract

Round 2

Reviewer 1 Report

Thank you for the corrections. The manuscript has been substantially improved with the modifications added to the text.

I only have a suggestion. I am not sure that the term encapsulation in biometic nanoparticles is correctly used in the manuscript that refers to enclose something in, like a capsule. This fact will depend on the extract properties and the water solubility or the lipophilicity of the compounds forming the extract in the lipid environment. I would suggest referring to these extracts as Liposomes enriched with Prunus Spinosa extracts better than biomimetic nanoparticles. The name nanoparticles is associated with a nanometer size of the particles that in this case might not be accomplished, and probably in this case the extracts will be embedded in micelles.

Author Response

We are thankful to the reviewer for this valuable comment, which allows us to clarify this point that we most likely missed to explain in our manuscript. The formulation described in this manuscript relies on two different steps:

  1. the biomimetic step is referred to the incorporation of membrane proteins purified from immune cells into the lipid bilayer of a liposome-like nanovesicle, termed leukosome, according to the protocols described in Molinaro et al. Nature Materials, 2016.
  2. as reported in the method section of the revised manuscript, the freeze-dried Prunus Spinosa Fruit (PSF) extract was first dissolved in ethanol, then added to the lipid phase, which was evaporated in a thin film and hydrated to form Leukosomes loaded with the PSF extract.

In the final formulation, we hypothesize that the PSF extract resides in the lipid bilayer, together with the hydrophobic domains of the membrane proteins. We agree with the reviewer that the term 'encapsulated' does not really fit with this scenario, since the PSF extract is not into the lipid core of the biomimetic nanoparticles, but most likely associated with the lipid bilayer. We used this term in its figurative meaning of something 'encapsulated' into the lipid bilayer. As a matter of fact, the DLS analysis reveals the presence of 140 nm nanovesicles for the loaded leukosomes group.

Therefore, to address reviewer's comment, we will use reviewer's suggestion to refer to biomimetic nanoparticles encapsulating the PSF exctract as 'Leukosomes enriched with Prunus Spinosa extracts'.

Reviewer 2 Report

The authors responded to all my comments and in the reviewer's opinion, the paper is publishable.

Author Response

we are very thankful to the reviewer for this comment.